# Moisture Computing-Based Internet of Vehicles (IoV) Architecture for Smart Cities

**DOI:** 10.3390/s21113785

**Published:** 2021-05-30

**Authors:** Ali Tufail, Abdallah Namoun, Adnan Ahmed Abi Sen, Ki-Hyung Kim, Ahmed Alrehaili, Arshad Ali

**Affiliations:** 1Faculty of Computer and Information Systems, Islamic University of Madinah, Madinah 42351, Saudi Arabia; a.namoun@iu.edu.sa (A.N.); adnanmnm@hotmail.com (A.A.A.S.); alrehailiium@iu.edu.sa (A.A.); a.ali@iu.edu.sa (A.A.); 2Department of Cyber Security, Ajou University, Suwon 16499, Korea; kkim86@ajou.ac.kr

**Keywords:** smart vehicles, Internet of Vehicles, IoV, sensors, cloud computing, MEC, IoT, smart cities, fog computing

## Abstract

Recently, the concept of combining ‘things’ on the Internet to provide various services has gained tremendous momentum. Such a concept has also impacted the automotive industry, giving rise to the Internet of Vehicles (IoV). IoV enables Internet connectivity and communication between smart vehicles and other devices on the network. Shifting the computing towards the edge of the network reduces communication delays and provides various services instantly. However, both distributed (i.e., edge computing) and central computing (i.e., cloud computing) architectures suffer from several inherent issues, such as high latency, high infrastructure cost, and performance degradation. We propose a novel concept of computation, which we call moisture computing (MC) to be deployed slightly away from the edge of the network but below the cloud infrastructure. The MC-based IoV architecture can be used to assist smart vehicles in collaborating to solve traffic monitoring, road safety, and management issues. Moreover, the MC can be used to dispatch emergency and roadside assistance in case of incidents and accidents. In contrast to the cloud which covers a broader area, the MC provides smart vehicles with critical information with fewer delays. We argue that the MC can help reduce infrastructure costs efficiently since it requires a medium-scale data center with moderate resources to cover a wider area compared to small-scale data centers in edge computing and large-scale data centers in cloud computing. We performed mathematical analyses to demonstrate that the MC reduces network delays and enhances the response time in contrast to the edge and cloud infrastructure. Moreover, we present a simulation-based implementation to evaluate the computational performance of the MC. Our simulation results show that the total processing time (computation delay and communication delay) is optimized, and delays are minimized in the MC as apposed to the traditional approaches.

## 1. Introduction

Our dependency on the Internet has increased significantly over the last two decades. This transformative technology has empowered accessibility to information and a myriad of services with a few clicks. Moreover, the Internet of Things (aka IoT) has taken the concept of Internet to another level where “things” or physical objects are interconnected within a communication network in which data are transferred seamlessly between these things. The so-called things can be thought of as various hardware technologies, e.g., sensors, embedded systems, and wearable devices, and software technologies, e.g., applications and real-time analytics [1]. The interconnection of the things enables collecting and analyzing diverse data to deliver information and services in a more contextualized and efficient way for the people. The actual applications of IoT can be observed in multiple fields; for instance, the whole process of diagnosis can be performed remotely, and home automation can be realized to create smart homes [2].

Humans are highly reliant on vehicles for their day-to-day commuting and transportation of goods and services. The Internet revolution has impacted the automobile industry by shifting the focus on realizing the concept and theories related to smart cars and intelligent vehicular communication based on IoT. The introduction of the Internet and related infrastructure into vehicles has evolved the vehicular ad hoc network (i.e., VANET) into the newly emerged concept of Internet of Vehicles (i.e., IoV). In effect, IoV deploys sensors to collect data related to several traffic-related events (like congestion, accidents, weather information, etc.) by utilizing diverse heterogenous networks [3]. Advanced sensors enable vehicles to sense, communicate, report, and react to the surrounding environment to achieve multiple benefits for the drivers, commuters, other vehicles, and law enforcement authorities. The main services that profit directly from such a concept include, but are not limited to, roadside assistance, emergency services, efficient navigation, traffic management and monitoring, traffic congestion reduction, pollution control, etc. [4]. 

IoV-based technology is finding wider acceptability among consumers and is making its way into the industry. The availability of high-speed Internet and related infrastructure increased the demand by drivers to stay connected while commuting. Drivers anticipate staying updated about the events in their surroundings to empower them to make smart driving decisions. According to McKinsey & Company Global Institute [5], the economic value of the IoT-based automotive industry is expected to range between $3.9 trillion and $11.1 trillion and the value of the intelligent vehicles sector is predicted to reach approximately between $210 billion and $740 billion by the year 2025 [5].

Several renowned companies like Apple, Google, and Nvidia have started to implement IoV technologies so that they can tailor-make their applications-specific integrated chips to fulfill specific requirements, such as the deployment of cutting-edge sensors, latest technologies for displays, real-time computing capabilities, provision of in-vehicle operating systems, artificial intelligence, and machine learning support for decision making etc. [4]. IoV-enabled vehicles bring about several benefits to business processes such as automation, supply chain management (SCM), and logistics. For example, businesses can track several services such as productivity, safety, fuel consumption, and rules compliance, using connected vehicles. Moreover, deliveries could be tracked easily throughout each phase (Raconteur-Content for business decisions). 

IoV services can be provided using the vehicular cloud computing paradigm [6], which delivers dynamic applications to commuters while on the go. For instance, travelers may be informed about real-time traffic information (e.g., incidents, delays, jams). However, IoV is a fast-evolving field, so the frameworks and protocols related to this domain are constantly changing. One of the critical factors to supplying effective IoV-based services is latency. Typically, the cloud computing paradigm plays a vital role in providing such services. Cloud computing is the traditional approach that implements a central infrastructure for fast processing. However, its distance from the end-user devices inflicts additional delays [7]. To combat this drawback, several works attempted to shift the computation towards the end-user to reduce the response time and quickly deliver the critical services. Concepts like Fog Computing (i.e., FC), Mobile Edge Computing (i.e., MEC), and Cloudlets were introduced, where processing is performed close to the user equipment (i.e., UE) [8]. All of these computing paradigms operate based on the concept of distributed or decentralized computing. However, there is a delicate balance between the proximity of computation with the UE and the cost and other management issues related to the distributed computing. Although cloud computing provides much-needed infrastructure and can help save costs, it inflicts additional network latency. Fog/mobile edge computing (i.e., FEC, MEC) reduces the latency but implicates cost, distributed resource management, reliability, congestion control, performance degradation for dense networks, and mobility issues [9]. 

Several attempts were made to deploy a hybrid architecture with the combination of IoV/Fog and software-defined network (i.e., SDN)/network function virtualization (i.e., NFV) to improve QoS for IoV. With the help of network slicing, requirements for different applications requiring varying bandwidth can be fulfilled, whereas task offloading to fog servers can reduce the latency significantly [10]. However, it is still an evolving field, and the complexity of appropriate resource allocation, network slicing, and task offloading are open research challenges [11].

Our motivation behind our work is flared by various research gaps, particularly:**Gap One** [12]: Cloud computing infrastructure placed at a long distance may be tens of kilometers away from the end device and may incur high latency and slower response time.**Gap Two** [12]: Mobile edge computing infrastructure placed closer may be tens of meters away from the end device and incurs high infrastructure cost since more computing infrastructure points will be required to cover a particular geographical area.**Gap Three** [9]: Mobile edge computing infrastructure cannot support varying traffic density; latency increases manifold once the load increases on the limited computing infrastructure.**Gap Four** [11]: Although SDN/NFV and fog computing-based approaches provide better QoS requirements for IoV applications, task offloading to MEC servers can degrade performance with regards to latency, especially in dense networks.

To mitigate the above research gaps, our research proposes a novel computing paradigm, which we call moisture computing (aka MC). Moreover, we propose a new communication architecture comprising a mechanism for IoV that could achieve the following:(a)minimize latency overhead;(b)cover a wide geographical area to deliver the required information and services, such as collision or congestion updates, to the drivers promptly;(c)mitigate shortcomings of existing approaches, i.e., Cloud Computing (centralized computing) and Edge Computing (distributed computing), by placing the computing infrastructure at an appropriate distance from the UE to give better latency in various scenarios;(d)reduce the infrastructure-related cost through the deployment of a middle layer hardware.

Overall, the suggested architecture implements a semi-distributed computing paradigm to reduce end-to-end latency while improving the reliability of the information processed through computations and analytics closer to the UE. We coined the term ’moisture computing’ since we want to bring the necessary processing capabilities as close as possible to the users. The term ‘moisture’ is borrowed from meteorology, where moisture signifies the vapor that stays in the atmosphere, typically between the clouds and earth surface. Therefore, the computation infrastructure will be made available below cloud infrastructure but above the edge of the network. The moisture computing paradigm combines the strengths of the two existing paradigms, i.e., cloud computing, which is central, and edge computing, which is highly distributed. It is important to note that fog computing is a concept that suggests shifting the computation, storage, and other infrastructure towards the edge of the network and closer to the UE to enhance performance in various ways. Depending upon the proximity of the infrastructure to the UE, a unique name has been assigned to that computing paradigm, including MEC, cloudlet, and mist computing [13]. The concept of our proposed paradigm, i.e., the MC, supports the fog computing narrative of pushing the computation towards the UE; however, the infrastructure has been suggested to be placed a couple of hops away from the UE, unlike other fog-based paradigms such as MEC which is one hop away and mist where the computation is performed inside the UE [13]. 

In [14] authors present cloud computing-based architectures for mobile vehicles that typically have three layers, i.e., onboard, communication, and cloud computing. However, most of these architectures suffer from high latency and other issues, such as security and data management challenges [15], although latency issues have been solved to a certain extent in MEC due to closer proximity to the end devices. However, the performance degrades significantly due to the increasing number of end devices in a particular area served by a single MEC. Additionally, scalability is also an essential issue for MEC-enabled solutions [16]. 

In our architecture, once an event is detected by a smart vehicle (SV) it is instantly reported to the moisture computing layer (MCL) with the help of a roadside unit (i.e., RSU). The MCL performs all necessary processing instead of the RSU. Next, the resulting instructions are forwarded to all RSUs of the vicinity. The RSUs assume the responsibility of transferring these instructions to their SVs to empower them to take precautionary measures and decisions.

The remainder of the paper is divided into six sections. Section 2 reviews the latest works in the area of IoV architectures, such as mobile edge computing. Section 3 presents the concept of smart vehicles and their characteristics. Section 4 details the moisture computing architecture and highlights its key components. Section 5 presents a benchmark study to showcase the advantages of the MCL over existing IoV architectures. Section 6 presents a computational comparison of MC, MEC, and cloud architectures. Section 7 concludes the implications of the proposed architecture for the smart transportation field. 

## 2. Related Work

In this section, we summarize the previous works conducted in the relevant field of study. The focus is on exploring the latest studies presented by the research community in solving issues pertaining to the Internet of Vehicles. 

Authors in [17] state that the IoV is playing a crucial role in developing intelligent transportation systems for smart cities. However, they claim that the traditional routing protocol-based methods cannot be used for these delay-sensitive IoV applications. They propose to utilize MEC along with Software Defined Networks to enhance the performance of IoV-based communication. In order to minimize the delay, they use several techniques, including placement and optimization algorithm for controllers. However, they do not talk about the cost-effectiveness of their proposed solution nor do they test the behavior of their proposed solution in varying network densities.

In another study [10], authors argue that fog computing can solve latency-related issues that are prevalent in cloud computing. However, the distributed and complex fog computing network structure can serve as a bottleneck for the performance of IoV-based applications. Therefore, they propose a 5G IoV architecture based on SDN and fog computing. They claim that their proposed architecture can help to allocate heterogenous computing resources with a QoS guarantee.

In [18] authors talk about bringing flexibility and better network management using SDN and network function virtualization (NFV) in IoV-related scenarios. They claim that the use of SDN/NFV can help IoV in improving communication, caching, and computing capabilities. However, they state that to effectively utilize concepts related to MEC with the combination of SDN/NFV techniques, several challenges such as joint resource slicing and access control should be addressed. 

Authors in [19] talk about 5G and fog computing architectures from a security perspective. They propose a scheme based on ciphertext-policy attribute-based encryption. They also incorporate the concept of user revocation in order to support dynamicity in vehicles. With the help of analysis and experiments, they claim that their proposed scheme is robust and secure against various threats in the IoV environment. 

In another study [20], authors discuss multimedia collection in IoV environments. They argue that the use of MEC servers can negatively impact the performance because of the unpredictable nature of the IoV network. Therefore, they present a traffic flow prediction-based resource reservation method. They deploy a deep spatiotemporal residual network to predict the traffic flow. They claim that their proposed method improves the performance manifold. 

Authors in [21] propose an IoV environment based on Intelligent Transportation System (ITS) big data analytics. The proposed architecture merged three dimensions, i.e., intelligent computing, real-time big data analytics, and IoV. The paper also discusses the IoV environment, ITS big data, and lambda architecture for real-time big data analytics. Similarly, [22] present CAVDO, which is based on a dragonfly clustering algorithm that used cluster-based packet route optimization to create stable topology and mobility aware dynamic transmission range algorithm (MA-DTR). The authors claimed that the proposed algorithm performed better than many other algorithms in many cases, provided a current cannel condition.

Authors in [23] talk about integrating mobile edge computing (MEC) in a vehicular environment with existing architecture to propose the next generation of ITS in smart cities. Their proposed protocol comprises three layers and can be used for traffic safety and travel convenience applications in the vehicular network. The authors claimed that the proposed protocol could help in improving data dissemination in terms of data delivery, delay, and communication overhead.

Authors in [24] propose a heterogeneous IoV architecture with a combination of multiple wireless interfaces exploiting the long WiFi and 4G/LTE installed on–board smart vehicles. The proposed architecture employed the Best Interface Selection (BIS) algorithm to select the best interface from available wireless interfaces to support connectivity for transmitting data efficiently in a vehicle to infrastructure (V2I) communication system. The authors claimed that the proposed algorithm performed best in an IoV environment to handle various types of applications against single wireless technologies.

Authors in [25] explore the agent-based model of information sharing in smart vehicles. Their focus was on the Social Internet of Vehicles (SIoV). They claim that their simulations proved that closure of social ties and its timing impacts the scattering of novel information significantly.

Authors in [26] propose a vehicular edge multi-access network to construct cooperative and distributed computing architecture for IoV. They proposed mechanism-based collaborative task offloading and output transmission to ensure low latency and application-level performance. The authors claim that the proposed scheme is capable of decreasing reaction time while guaranteeing application-level driving experiences. In yet another study [27], authors propose IoV architecture and protocol stack for cooperative communication. The study aims to explore the option of providing trustworthy services according to user requirements. They also test their proposed location-based method via extensive experimentation. 

Authors in [28] propose a model called Intelligent Vehicular Sensor Network (IVSN) by using spatial, data, context, and group awareness for the Internet of Vehicles (IoV). They claimed that the proposed model fully utilized raw data collected from vehicle sensors. To understand the model roles and interaction between different Sensing as a Service (SaaS) entities were defined. Moreover, performance evaluation of the proposed model was done using real data obtained from specific sensors in the given vehicles.

Table 1 shows the results of comparing our proposed architecture with the most relevant literature, based on the following criteria:Distributed/Central/Hybrid: The selected architecture can be central, distributed, or hybrid.Type of Approach: Several approaches can be used to perform computations such as cloud computing, fog computing, mobile edge computing, etc.Delay/Latency: It is crucial for the architectures to talk about the expected delay or latency that might incur using a particular architecture.Domain of Application: Several applications can be proposed based on utilized architecture. These applications can be related to safety, security, emergency services, convenience, immersive experience, etc.Smart City Support: The architecture should have the capacity to support various applications and scenarios of a smart city.Cost-Effectiveness: Infrastructure-based cost is an important criterion to consider before deploying any particular architecture.Density: Density has an influential impact on the latency and response time of a particular architecture. In a dense network, there will be more strain on the computing infrastructure.Simulation/Analysis: This criterion refers to the validation of the proposed architectures. The validation can be done using mathematical analysis, simulation, or real-time implementation.

Although IoV promises revolutionary changes for transportation processes, it opens a wide array of issues that need to be solved. The prominent issues in this field are highlighted below [29,30]:Lack of a Common Standard

Currently, there exists no common standard for the IoV industry. It creates problems for effective and efficient communication amongst various critical components of an IoV network, such as V2V communication, V2R communication, V2I communication, etc. Various big companies and institutions will need to join efforts in long-term partnerships to create a robust and open standard that would cover all aspects of IoV communication. 

2.Big Data

IoV is expected to generate a lot of traffic, and each node (vehicle) will be expected to process more than a gigabyte of information per second. This will be a huge challenge to provide efficient information processing and timely arrival of valuable data and information. The presence of cloud, edge, and fog computing can play an essential role in giving a solution to this issue. The introduction of 5G technology is also going to play a pivotal role. 

3.Privacy and Security-Related Issues

One of the biggest challenges in IoV is security and privacy. Since many nodes will be making themselves a part of a complex and integrated network, there would always be a chance of leaking personal information. Many vulnerabilities could be also explored by adversaries to launch several kinds of attacks. Strong authentication and authorization protocols and encryption techniques can help to resolve this issue. 

4.Mobility and Reliability

These two issues are of significant importance. Since many applications will be critical and sensitive, they would require highly reliable and timely information. The mobile nature of nodes makes it more complex and can reduce the reliability of the information. Mobile technologies like 5G and better communication protocols can help to solve these issues in a IoV network.

5.SDN Resource Management

The role of SDN for fog computing-related concepts is promising; however, it is still evolving, and there are issues such as resource management and orchestration that can hinder the performance. Since IoV is dynamic and scalable, so appropriate slicing is crucial to maintain strict performance requirements. Moreover, appropriate placement of edge computing infrastructure can also play a role in reducing the end-to-end latency. 

## 3. Smart Vehicles

The idea of smart vehicles is becoming a reality now. Several manufacturers are interested in designing a smart vehicle that would suit the everyday commute and make the whole journey a unique experience for the passengers. A typical smart vehicle will be equipped with the latest technological equipment to serve the needs of the passengers on the go. Figure 1 highlights the important technological components of a smart vehicle:

A smart vehicle should have the capability to communicate with a variety of nodes. For this, it must be equipped with a heterogeneous set of technologies. It must allow the passengers to communicate with the help of WiFi, cellular technologies, Bluetooth, etc. The smart vehicle should also be able to communicate with the satellite for precise navigation services. Another important component of the smart vehicle is the sensor and actuators. Each smart vehicle should be equipped with a wide range of sensors, for example, in order to monitor the road conditions, temperature, humidity, fire, etc. The Machine to Machine (M2M) communication capability should be inbuilt in these smart vehicles where they should communicate with other nodes without human intervention, for example, in case of an accident or an emergency. 

Google was the first to introduce a smart vehicle (i.e., driverless car) in the year 2010. Although smart vehicles are not very common at the moment, by 2025 the global market size of smart cars will be 166 billion USD [31]. 

It is expected that these smart vehicles will enhance the safety of passengers on the roads. With the help of onboard smart devices, a smart vehicle can send the crash site to the emergency services. Moreover, it can broadcast crash alerts to other vehicles heading towards the crash site. It can help to save time and, most notably, many lives. The IoV is evolving fast, and with the launch of 5G, it is expected that smart vehicles will be seen on the road in huge numbers with a fancy set of applications for passengers. 

A smart vehicle can be involved in various ways of communication. Therefore, we have summarized some of the communication models below [32]:V2V: This type of communication takes place between vehicle to vehicle directly.V2I: This type of communication takes place between vehicle and road side infrastructure.V2R: This type of communication takes place between a vehicle and roadside unit.V2N: This type of communication takes place between a vehicle and network.V2P: This type of communication takes place between a vehicle and pedestrian.V2D: This type of communication takes place between a vehicle and device.V2G: This type of communication takes place between a vehicle and gateway.V2X: This shows a generic way for showing communication between a vehicle and any other entity

Smart vehicles can support a wide array of applications that will be beneficial in assisting drivers by providing real-time traffic and road condition information and providing an immersive experience for all travelers by providing entertainment and other services on the go. Typically, we can broadly divide IoV applications into the following categories [29]:On-road safety applications;Entertainment and onboard passenger applications;Comfort and convenience-related applications;Traffic management applications;Emergency services.

## 4. The Proposed Moisture Computing IoV Architecture

Figure 2 depicts the proposed moisture-computing IoV architecture, which exhibits a heterogeneous communication network comprising multiple technologies. Smart vehicles can communicate directly with other vehicles (V2V), pedestrians (V2P), and cellular phone infrastructure (V2I). On the other hand, the roadside units can communicate with each other (R2R) and the nearby vehicles (V2R). 

Overall, the main components of our proposed MC architecture include:Smart Vehicles (denoted as SVs)Roadside Units (denoted as RSUs)Base Stations (denoted as BSs)The Moisture Computing Layer (denoted as MCL)The Central Information Processing Center (denoted as CIPC)/CloudComputing Points

### 4.1. Smart Vehicles (SVs)

SVs are the architecture building blocks since they represent mobile nodes, which interact with each other and with the other architecture components like RSUs or BSs. We assume that SVs have the following capabilities:•SVs are equipped with multiple wireless interfaces to support communication using various standards, such as WAVE, WiMAX, and cellular communication like 4G/LTE/5G [33];•SVs are empowered to carry out V2V, V2I/R, and V2P communication;•SVs can indirectly communicate with the cloud to access the Internet and other related services;•The central role of RSUs is to receive and relay the information coming from or going to SVs;•Furthermore, the GPS provided within SVs will help locate the exact location of the incidents easily, and aid can be dispatched quickly.

### 4.2. Road Side Units (RSUs)

RSUs are fixed nodes deployed by the roadside. We assume that RSUs have the following capabilities:•RSUs can send and receive messages on different frequencies from the SVs within the transmission range;•RSUs connect to the MCL via a reliable link, preferably using an optical fiber;•RSUs operate as relay nodes that receive and forward messages;•RSUs are resource-constrained; therefore, information processing is carried out at the MC layer, which comprises rich computational resources. In contrast to the cloud and edge computing, computation power is placed in the middle of the architecture;•The RSUs communicate with the SVs and other RSUs using WAVE/IEEE 802.11p [34].

### 4.3. Base Stations (BSs)

BSs are fixed nodes used to provide a myriad of services to the smart vehicles directly without the involvement of RSUs using cellular communication technologies (e.g., 4G/LTE/5G). When a 5G Internet connection is used, bandwidth-consuming services (e.g., entertainment) will be made accessible on the go. BSs are also directly connected to the MCL. The existence of cellular communication infrastructure offers flexibility to the SVs to choose from multiple interfaces depending upon the SVs’ requirements and quality of service (i.e., QoS) factors, such as location, availability, latency, among others. Typically, BSs are connected to their Mobile Switching Centers to carry out their cellular network-based communication. 

### 4.4. Moisture Computing Layer

The MC layer is responsible for fulfilling the demands of a specific geographical area. Based on the SVs requirements, e.g., the density of SVs or urban/rural settings, the computing points can be decided to support the MCL. Each computing point contains the necessary infrastructure to perform computations at the MCL. Instead of having just one CIPC, various tasks are delegated to these MCL to decrease the response time and communication latency. This approach is expected to significantly reduce the required infrastructure compared to deploying the infrastructure at the edge of the network (i.e., the case of MEC). 

### 4.5. The Central Information Processing Center (CIPC)/Cloud

CIPC manages all computing points of the MCL. CIPC possesses high computation and storage capabilities and thereby supports a wide range of services and applications. In an urban environment, various algorithms and analytic tools can be utilized to offer drivers several services such as route planning and guidance, real-time traffic congestion information, parking and shopping information, etc.

### 4.6. Computing Point

We define computing point as the location that contains the necessary computing infrastructure. In our proposed MC architecture, the computing point is at the MCL. Figure 2 shows the MCL location below the cloud infrastructure and towards the edge of the network. 

Table 2 sums up the infrastructure required within an urban geographical area (e.g., 2 square kilometers) to release our MC architecture.

In effect, our MC IoV-based communication comprises three layers, where each layer is responsible for a different set of tasks, as depicted in Figure 3. This setup will help minimize the complexity and facilitate the management of the overall network. 

### 4.7. Sensing and Reporting Layer

The bottom layer, also called the sensing and reporting layer, contains mainly the mobile SVs. These SVs are equipped with state-of-the-art technologies to conduct V2V, V2I, and V2R communication. The SVs are supposed to be in considerable numbers depending on the roads’ capacity; however, they are resource-constrained entities with limited storage and processing capabilities. SVs will be involved in sensing and reporting events. Depending on the type of the event sensed, SVs will communicate with (1) RSUs or (2) RSUs and other SVs in their vicinity. SVs can select the desired communication interface based on the detected event or user request. Figure 4 illustrates an example scenario where an accident has occurred and subsequently been reported by both SVs’ onboard sensors. The figure shows the flow of information to and from the SVs as uplink and downlink communication. The bottom layer directly communicates with the other SVs and the middle layer infrastructure. 

Algorithim 1 explains the pseudocode for the sensing and reporting process for each SV. Upon detecting the event, it will follow the given process and will report the event. Depending on the type of the event, RSUs and the MCL will take the required actions.
**Algorithim 1** Sensing and Reporting for smart vehicles 1: on_sensing_an_event(sensed event)
 2: event = sensed event 
 3: message = event, type, location, speed,…… 
 4: **if**(event_emergency) 
 5: **check**(RSU_connection) 
 6:  **if**(RSU_connected) 
 7:   send(message_to_RSU) 
 8:   send(message_to_SVs)
 9: **else if**(BS_connected) 
 10: send(message_to_BS) 
 11: send(message_to_SVs) 
 12:  **else**

 13: scan_to_connect 
 14: **if**(connected) 
 15:  on_sensing_an_event(event) 
 16: **else**

 17: **check**(RSU_connection) 
 18:  **if**(RSU_connected) 
 19:   send(message_to_RSU)
 20: **else if**(BS_connected) 
 21:   send(message_to_BS) 
 22: **else**

 23:   scan_to_connect 
 24:   **if**(connected)
 25:  on_sensing_an_event(event)

### 4.8. Processing Layer

The middle layer, also called the processing layer, contains the infrastructure that carries out communication in the proposed IoV network. This layer coordinates the functions of components residing within the top and bottom layers. The major components of this layer are RSUs, BSs, and the MCL. RSUs will connect to SVs and communicate using a wireless access technology like WAVE/IEEE 802.11p [33]. The communication between SVs and RSUs is bidirectional and involves information sharing or information request messages from the SVs to the RSUs and vice-versa.

Moreover, SVs can use other interfaces like 3G/4G/5G in specific scenarios. The interface selection will depend on the available signal strength, the application requirement, and the type of information required [32]. In our proposed architecture, the SVs can connect to the nearest available BS, which is connected to the MCL and mobile switching center. 

The MCL in this layer acts as the backbone connecting the roadside infrastructure with the Internet, cloud, or mobile switching centers. Each computing point in the MCL manages the BSs and RSUs of a particular geographical region. 

Figure 5 shows the flow of information and interaction between sensing, reporting and processing layers of our propsed MC architecture. It also distinctly defines the flow of information for emergency and non-emergency events.

### 4.9. Cloud/Internet Layer

The top layer, also called the Cloud/Internet layer, provides Internet connectivity, data storage, data processing, and other critical services. CIPC possesses high computation and storage capabilities to support a wide range of services and applications. Analytics tools can be deployed here to offer varied services to the SVs. The smart services range from route determination and guidance, total journey time estimation, traffic congestion information, parking, and shopping information.

The proposed MC architecture supports the notion of fog computing, where the computing, storage, and other infrastructure is pushed towards the edge of the network or closer to the UE. We propose to place computing points close to the UE but still not one hop away like the MEC or tens of hops away like in the cloud. This suggested proximity of computing infrastructure will impact the IoV communication in the following ways:The end-to-end latency will be reduced compared to the cloud since the processing will be done much closer to the SVs.SVs will get a faster response in a dense network since MEC cannot cater to a high number of requests due to limited computing infrastructure.MC infrastructure is expected to cover a wider area and eventually require lesser computing points reducing the overall infrastructure cost in comparison to the MEC.

## 5. Performance Analysis and Discussion

In this section, we perform a series of analyses and discuss different scenarios, which could be faced by the SVs while on the road in a smart city. With the help of mathematical modeling, we predict the behavior of our proposed MC architecture. Table 3 lists the key criteria of cloud computing, MEC, and MC architectures based on the previous works [8,33]. 

A vital factor in saving lives, time, and resources on roads is to promptly provide traffic news and guidelines to drivers to enable them to make appropriate decisions that avoid potentially disastrous consequences. With the help of semi-distributed computing, our proposed architecture aims to deliver this critical information to the drivers reliably. 

Figure 6 shows a potential scenario of an urban/semi-urban setting. For simplicity reasons, just the SVs and the RSUs are depicted in Figure 6. The approximated ranges of both SVs and RSUs have been indicated as well. Although the SVs can employ one-hop communication to send information to other SVs, the information cannot be distributed to a wide area due to their limited communication range. However, our architecture uses the resources almost at the edge of the network to process and distribute information to a large geographical area. The MCL uses all of the RSUs of an area simultaneously to propagate information to the drivers and affected SV(s).

We compared our approach, i.e., MC, with two renowned approaches, namely (1) cloud computing and (2) mobile edge computing (MEC). Cloud equipment is usually placed approximately tens of kilometers away from the user equipment, whereas MEC nodes are typically placed tens of meters away from the user equipment [35]. However, the MCL will be located hundreds of meters away from the end-user devices. To achieve a reasonable comparison, we benchmarked our results against the findings reported in [36] and [37]. Table 4 highlights various analysis parameters that are derived from the benchmarked studies. 

### 5.1. Round Trip Delay Estimation

In this section, we estimate the time required to send the emergency information and share it with the other SVs in a particular area to notify drivers about incidents and delays that could be averted. In our architecture, the affected SVs report the events to the nearest (connected) RSU and broadcast it to its first-hop neighboring SVs. The RSU will forward the event to the MCL, which in turn forwards the instructions to all RSUs of that specific area. Finally, the RSUS will notify its SVs instantly to enable them to make crucial decisions.

Three prominent cases are considered in this comparison. The first case estimates the round-trip delay in the MEC, where the limited infrastructure is placed next to the RSUs to carry out the local processing. The second case estimates the round-trip time in the MCL. The third case estimates the round-trip delay in cloud computing. Subsequently, we compare the three cases and perform the analysis. Below we use the terms delay and round-trip delay interchangeably. Formally, we define round-trip delay as “the amount of time it takes an SV to report an event to the connected RSU and to receive the relevant notifications or instructions from the RSU. This also includes processing and forwarding time at RSU and beyond.”

#### 5.1.1. Case One: MEC Based Approach

In this case, the MEC-based approach places limited infrastructure next to the RSUs, typically placed tens of meters away, to provide necessary information to the SVs in the range. Examples of such information can be related to local weather forecasts or traffic congestions in the local area. RSUs receive and forward the instructions to and from the nearest SVs. Information flow will be SVs→RSU→(SVs and RSUs). Consider the following equation for this case:*T_tr_* = *T_vr_* + *T_i1r_*(1)
*T_i1r_* = *µ_r_* + *γ_r_*(2)
where *T_tr_* represents the total estimated time to receive, process, and disperse the needed information without involving resourceful MCL or cloud infrastructure, *T_vr_* represents the time it takes the affected SV to sense and report the event to the nearest RSU and the one-hop SVs using a broadcast message, *T_i1r_* represents the time taken by an RSU to receive and process the data and send the information to the nearest SVs and one-hop neighboring RSUs. The amount of time it will take at an RSU depends on two factors, i.e., processing delay, which we call *µ_r,_* and the propagation delay, which we call *γ_r_*. We assume that the forwarding time is negligible since one broadcast message can reach all one-hop connected neighbors. In this scenario, the efficiency of the RSU will depend on the density of neighboring RSUs and the number of SVs connected to the RSU. Adding more neighboring RSUs and the SVs will inflict more load on the connected RSU since it will need to forward messages and process them. Moreover, if more RSUs are installed in the range, a particular RSU will receive more messages directly from those RSUs, thus increasing the queuing delay. Hence, it is fair to assume that there might be variable delays in processing and responding to a particular request in case one. 

*µ_r_* will depend on the density of SVs, which we call β, and the density of RSUs, which we call δ.

The higher the number of RSUs and the SVs in a particular area, the longer the delay at the connected RSU because it will be involved in more forwarding and processing of events and instructions. Therefore, we can write: *µ_r_* = T_SV_ + T_Rsu_(3)
where T_SV_ is the time taken by a particular RSU to serve the connected SVs, and T_Rsu_ is the time taken by a particular RSU to serve the one-hop connected RSUs. We can deduce that: T_SV_ ∝ β(4)
T_Rsu_ ∝ δ(5)

We can introduce the constants *c* and *d* for the relationship defined above as follows:T_SV_ = c β(6)
T_Rsu_ = d δ(7)
where c and d represent the time spent to process a message at a particular RSU for a single SV and a single RSU, respectively. The higher the density of SVs or RSUs in a particular area, the longer the latency and delay.

We can therefore rewrite Equation (1) as follows:*T_tr_ = T_vr_* + (c β + d δ + *γ_r_*)(8)

#### 5.1.2. Case Two: The MCL-Based Approach

In this case, we consider our proposed architecture, i.e., the MCL. We assume that a medium level of computing infrastructure is placed not far, typically hundreds of meters away from the end-user devices (in this case, the SVs). Information flow will take the path SV→RSU and RSU→MCL→RSUs→SVs. Consider the following equations: *T_tm_ = T_vm_ + T_i1m_ + T_rm_*(9)
*T_i1m_ = µ_m_ + γ_m_*(10)
*T_rm_ = µ_mm_ + 2γ_mm_*(11)
where *T_tm_* represents the total estimated time where the MCL performs the computation and disperses the information, *T_vm_* represents the time it takes the affected SV to sense and report the event to the nearest RSU and one-hop SVs with the help of a broadcast message, *T_i1m_* represents the time it takes an RSU to receive and forward data to the MCL. It is worth noting that the RSU time depends on two factors, i.e., processing delay, which we call *µ_m,_* and propagation delay, which we call *γ_m_*. In this case, we assume that the only functionality of the RSU is to receive and forward the messages to and from the SVs. Therefore, we can assume that the processing delay is negligible with minimal queuing time. However, the processing delay time might increase slightly depending on the number of connected SVs. We can assume the following equation: *µ_m_* = f ρ(12)
where ‘f’ represents the time it takes to process a single message at an RSU for a single SV, and ρ represents the density of SVs in a specific neighborhood. More SVs in a particular area will result in longer delays. We assume that all communication occurs via the MCL instead of RSU–to–RSU communication. 

*T_rm_*, in Equation (9), refers to the time taken by the MCL to process and send the data to all RSUs of the affected area. The amount of time it takes to process data at MCL will be much less than that of the RSU because of the available resources. This processing time will also depend on the connected RSUs and SVs; for example, connecting more RSUs and SVs is likely to produce more queries for the cloud infrastructure to handle, eventually increasing the processing delay at the MCL. For comparison purposes, the number of SVs is considered for one RSU that is connected to the affected SV. Therefore, we can write:*µ_mm_* = e (υ_r_ + υ_s)_(13)
where ‘e’ represents the time needed to process a single message at the MCL for a single RSU, υ_r_ represents the density of RSUs, and υ_s_ represents the density of SVs. In Equation (11), we assume double the propagation time to cover the total round-trip time. 

The MCL can send messages to the emergency response services to provide on-road assistance to the affected SVs on time. Equation (9) can be rewritten as follows:*T_tm_* = *T_vm_*+ f ρ + *γ_m_* + e (υ_r_ + υ_s)_ + 2*γ_mm_*(14)

#### 5.1.3. Case Three: Cloud-Based Approach

This case considers the involvement of cloud infrastructure, where only the cloud processes and distributes the information to the RSUs and SVs. The cloud is assumed to have an appropriate computation infrastructure placed far away from the SVs. 

Information flow in the cloud case is SV→RSU→Cloud→RSUs→SVs. Consider the following equations for this case:*T_tc_* = *T_vc_ + T_i1c_ + T_rc_*(15)
*T_i1c_ = µ_c_ + γ_c_*(16)
*T_rc_ = µ_cc_ + 2γ_cc_*(17)
where *T_tc_* represents the total estimated delay to receive, process, and disperse the needed information by the resourceful cloud infrastructure but without the MCL, *T_vc_* represents the time it takes the affected SV to sense and report the event to the nearest RSU and one-hop SVs using a broadcast message, *T_i1c_* represents the time it takes an RSU to receive and send the data to the cloud infrastructure. As stated previously, the RSU time depends on two factors, i.e., the processing delay (i.e., *µ_c_*) and propagation delay (i.e., *γ_c_*). Similarly to case two, we assume that the only functionality of the RSU is to receive and forward the messages to and from the SVs. Therefore, the processing delay is negligible with minimal queuing time. However, the processing delay might fluctuate slightly depending on the number of connected SVs. We can assume the following equation: *µ_c_* = g σ(18)
where ‘g’ represents the time taken by an RSU to process a single message for a single SV and ‘σ’ represents the density of SVs. 

*T_rc_*, in Equation (15), is the time spent by the cloud infrastructure to process and send the data to all RSUs of the affected area. It is important to note that the information processing time taken by the cloud infrastructure will be less than that of the MCL and RSUs. However, the propagation delay will be greater than both the MCL and RSU-based approaches due to the infrastructure placement in the network architecture, i.e., far away from the RSUs and SVs. Equation (17) highlights that the total delay at the cloud will depend on the processing delay (*µ_cc_*) and propagation delay (*γ_cc_*). The processing delay will depend on several factors, including the load on the cloud infrastructure and the traffic density. Traffic density is measured by the number of connected SVs as well as connected RSUs. Therefore, we can write:*µ_cc_* = h (υ_r_ + υ_s)_(19)
where ‘h’ is the time required by the cloud to process a single message for a single RSU, υ_r_ is the density of RSUs, and υ_s_ is the density of SVs. In Equation (17), we consider doubling the propagation time to cover the total round-trip time, since we measure the total time it will take the message to reach the cloud and come back to the SVs. 

The cloud can also send messages to the emergency response services to provide on-road assistance to the disaster-stricken SVs on time.

Therefore, Equation (15) can be rewritten as follows:*T_tc_* = *T_vc_* + g σ + *γ_c_* + h (υ_r_ + υ_s)_ + 2*γ_cc_*(20)

Upon inspection of Equations (1), (9), and (15), one might think that *T_tm_* is greater than *T_tr_* and *T_tc._* However, the following observations are worth noting:The time *T_vr_*, *T_vm_,* and *T_vc_* will be similar since the affected SV will take the same amount of time to sense and report the event to the nearest RSU in all three cases.The time *T_i1m_* and *T_i1c_* will be the same; however, these two times will differ from *T_i1r_* and will vary significantly. This is because the RSU’s main job in the second and third cases will be to receive and forward the data, unlike the first case, where they will be responsible for processing and forwarding the information to first-hop neighboring RSUs. The RSU–to–MCL and RSU–to–cloud connections will use a high-speed transmission medium, such as optical fiber; however, RSU–to–RSU communications will be slower and less reliable since they will rely on a wireless communication medium. Moreover, the delay will be significantly longer than the other two cases (Equations (6) and (7)). Therefore, we can easily assume that,
*T_i1m_ << T_i1r_ & T_i1c_ << T_i1r_*(21)

Equation (21) signifies that *T_i1m_* and *T_i1c_* will be many orders smaller than *T_i1’_*.

In the first case, there is no MCL or cloud, so there will be no *T_rm_* or *T_rc_*. However, the MCL and cloud infrastructure are resource-rich in comparison to RSUs. Therefore, the time it will take to process data in the MCL and cloud infrastructure will be considerably smaller than the RSUs processing time. We can conclude from Equation (21) that removing *T_rm_* or *T_rc_* from the first case will not reduce the overall delay time. 

*T_rm_* will be smaller than *T_rc_* because of the added propagation delay. The cloud infrastructure is usually deployed away from the end devices, i.e., SVs and RSUs, compared to the MCL scenario. The deployment of the MCL will be towards the edge of the network so the propagation delay will be much less than that of the cloud infrastructure. Therefore, we can easily assume that:

*T_rm_ << T_rc_*(22)

Hence, considering Equations (21) and (22), it can be deduced that the time needed to send crucial instructions with the help of the MCL will be reduced in comparison to the cases where no MCL is used. Therefore, our proposed architecture can help deliver information quickly to the drivers to empower them to make appropriate decisions. We can assume that:*T_tm_* << *T_tc_* << *T_tr_*(23)

We compared three three computing paradigms, i.e., MEC, MC, and Cloud, and noted their behavior with a varying number of requests. Figure 7 shows that the denser the network, the higher the number of requests. With a smaller number of requests, the MEC performs better due to its proximity to the SVs. Since the MEC is just one hop away from the end devices and there are just a few requests, the end-to-end latency is reduced. However, as the number of requests increases, the performance of MEC infrastructure degrades significantly, which increases the latency. This degradation in the latency is due to the limited computing infrastructure of MEC, which cannot cope with the increasing number of requests where MEC infrastructure has to receive, process, and disperse the needed information. With the cloud infrastructure, due to added propagation latency, the overall latency is higher once the number of requests is low; however, with a more significant number of requests, the other two computing paradigms get constrained due to lack of computing infrastructure. Be mindful that cloud computing infrastructure performance also degrades after several requests increase a certain threshold [35]. Although cloud infrastructure has more computing resources than the other two paradigms, cloud infrastructure is expected to receive much more requests in peak hours since it covers a wider geographical area. This serves as the primary reason for cloud performance degradation, as MCL-based infrastructure outperforms the other two computing paradigms for less and a medium number of requests. 

Figure 8, Figure 9 and Figure 10 present the impact on latency with varying numbers of RSUs and SVs. The increase in the number of RSUs and SVs will have a different impact on all three computing paradigms. A fewer number of RSUs and SVs represent a sparse network so there will be less demand on the computing infrastructure while a more significant number of RSUs and SVs represent a dense network; therefore, the computing infrastructure is expected to suffer more strain on the resources in this case. Figure 8 shows the case where we change both the RSUs and SVs. Once the number of RSUs and the SVs are less, cloud computing shows the worst performance due to added propagation latency. However, MEC shows the least latency, but with the increase in the number of SVs and RSUs, the MEC performance degrades significantly. This decline in performance is due to the limited computing infrastructure, which cannot efficiently handle the high number of data handling and processing requests. The MCL performs much better than both MEC and the cloud in a less dense network, nevertheless with the increase in the density of the network, the infrastructure constraints increase the latency. 

Similar observations can be seen in Figure 9 and Figure 10. The density of the network is a major factor for an increase or decrease in latency. With an average density, which usually will be the case, the MCL outperforms the other computing paradigms since it reduces the latency significantly. The increase in the number of RSUs causes a surge in the number of messages and requests and puts more strain on the computing resources in all three paradigms. The increase in response time due to added processing is visible in MEC right away. However, this is evident in the MCL and cloud after a particular threshold since the MCL and cloud are rich in resources in comparison to the MEC. Therefore, we can conclude that the MCL provides an appropriate solution in terms of latency and cost by placing medium-scale computing infrastructure towards the edge of the network. 

## 6. Experimental Configuration and Evaluation

To validate our proposed approach, we opted to conduct a computational simulation, which was built using the C-Sharp programming language. The source code and executables of the simulation are made available and can be accessed at the GitHub repository (https://github.com/anamoun/mcl.git, accessed on 25 May 2021). In summary, we have compared the MCL architecture with MEC and Cloud computing architecture concerning three performance metrics, namely (1) computation time (CT), (2) communication delay (CD), and (3) total processing time (TPT). These three metrics should suffice to realize an efficiency assessment of the smart vehicular data transmission within the three architectures, specifically mobile edge computing (MEC), moisture computing layer (MCL), and cloud.

First, we will detail our computational assumptions for the three architectures. 

•The computational power of MEC, the MCL, and Cloud increases respectively, and this is expressed using the number of server nodes in each architecture.•We assume that each server node has the same technical specifications, leading to similar per task processing time among the nodes. In our comparison scenario, we presume that the processing time by a single node for each task of size 200 MB is 100 milliseconds. More significant tasks incur longer processing time.•The more nodes an architecture has, the faster the messages are processed and relayed back to the RSUs.•The propagation delay increases as we get further from the end-user, making the MEC the fastest and cloud the slowest to connect to the smart vehicles.•Traffic density (i.e., number of SVs occupying 1 km of roadway) can be expressed through the data generated in the roadways. More data infer that more messages are being generated from the vehicles and thereby higher traffic density.

In our simulated environment, data dissemination involves propagation of traffic conditions and event messages, including incidents, accidents, congestions, and road closures, among others. The propagated messages can realistically vary in size to accommodate a myriad of multimedia formats (simple text messages, images, videos, etc.). Other simulation parameters that were used for our comparison purposes are summarized in the below table. Table 5 reports four main aspects deemed to impact the computation delay, communication delay, and total processing time. These aspects cover the computational capacity of the three architectures, the processing delay inflicted by the three architectures, the propagation delay within the three architectures, and task information such as message size, packet size, and data generated by the smart vehicles.

The parameters outlined in Table 5 are configurable as depicted in the interface of our simulation tool (see Figure 11). We relied on the previous works and experimental setup to set our parameters, i.e., [36] and [37]. Other researchers may access the source code and implementation and tweak the parameters to test their scenarios as required. Once the computational parameters are set for the three architectures, the researcher should input the data generated from the SVs, as a series, to generate the charts. Our simulation provides the possibility to draw the charts instantly and export the results into an excel spreadsheet for further exploration and manipulation.

### Simulation Results

One of the critical performance metrics to measure the efficiency of the MCL approach is computation delay. This metric refers to the time taken by the nodes to complete a computational process (i.e., processing the messages received from the RSUs). Figure 12 was generated based on the assumption that each 200 MB task requires approximately 100 ms at the cloud, 200 ms at the MCL, and 800 ms at the MEC. Evidently, the MEC takes a significantly longer time to process the received messages. However, the MCL architecture provides acceptable results when compared to the cloud. 

Our second performance metric is communication delay which, in our case, refers to the latency taken by the messages to travel from the origin smart vehicles through the network (e.g., cloud) and back to the other smart vehicles. Figure 13 compares the communication delay between the three architectures. It shows that the communication delay is highest at the cloud, followed by the MCL and MEC. This result is anticipated since the MEC is closest to the smart vehicles. More events (i.e., events and data) cause significant communication delays for the cloud. 

The third important metric we investigated is the total processing time, which incorporates all delays involved from the point a traffic event is detected (Td) and sent, as a message, by a smart vehicle till the time (Tr) it is received by all smart vehicles in the smart environment. In other words, the total processing time represents the total dissemination delay considering the computation time and communication delay inflicted by the infrastructure. This chart is the most interesting graph in our comparison quest since it empowers us to draw a final verdict about the performance of our proposed architecture. Figure 14 shows that our approach outperforms MEC and Cloud. Although the difference between the MCL and MEC is subtle, we believe that adding more server nodes to the MCL will dramatically improve its overall performance in contrast to the MEC. Moreover, substantially bigger tasks (e.g., those involving videos, etc.) will result in significant delays in the MEC. 

## 7. Conclusions and Future Works

In this paper, we demonstrate the use of an efficient communication architecture (MC) to realize the concept of the Internet of Vehicles (i.e., IoV). To this end, a hierarchical architecture was proposed where various technologies are integrated to provide a number of services for the convenience of drivers and passengers. This architecture is also expected to assist in resolving the traffic monitoring and management issues. Unique characteristics of deploying the moisture computing infrastructure towards the edge of the network hierarchy have been presented. The MC tackles the drawbacks of having an infrastructure too close to the end devices, i.e., in the case of the MEC, and deploying an infrastructure too far from the end devices, i.e., in the case of cloud computing. Using mathematical analyses, we proved that the MC not only minimizes the latency in various scenarios but also reduces the infrastructure-related costs. Moreover, our simulation-based results showed the superiority of our MC approach over the MEC and cloud computing with respect to the total processing time. Overall, the MC reduces the communication delay suffered in cloud computing and improves the processing capabilities missing in the case of mobile edge computing. While our MC architecture provides promising results in the case of extensive data (i.e., in a dense network), it may not work so well when small data (i.e., in a sparse network) are generated in the network. In the event of small data, we expect the MEC to outperform our approach since the computation delay is reduced significantly. However, we believe that it is not uncommon for extensive data to be generated in an IoV network, such as when capturing HD videos or during the rush hours when a large number of smart vehicles request data simultaneously. On the other hand, the cloud provides significant processing capabilities but often results in unnecessary communication delays and possible misuse of its resources, especially when the tasks to process are small. For instance, most of the SVs need data and information on the fly to make smart and quick decisions on the roads.

The proposed architecture may be used to predict traffic congestions and dispatch emergency and roadside assistance. Future work will focus on testing our approach using variable settings regarding the messages size, traffic density, area size, number of RSUs, etc. We also aim to benchmark the computational results against the results of previous findings of other competing architectures. Additional tests will also involve simultaneous propagation of messages from different smart vehicles in the same area. Furthermore, we propose coordinating the three layers to dynamically select the best architecture to use based on the traffic density and size of generated data to reduce the communication delay in various smart city scenarios.

## Figures and Tables

**Figure 1 sensors-21-03785-f001:**
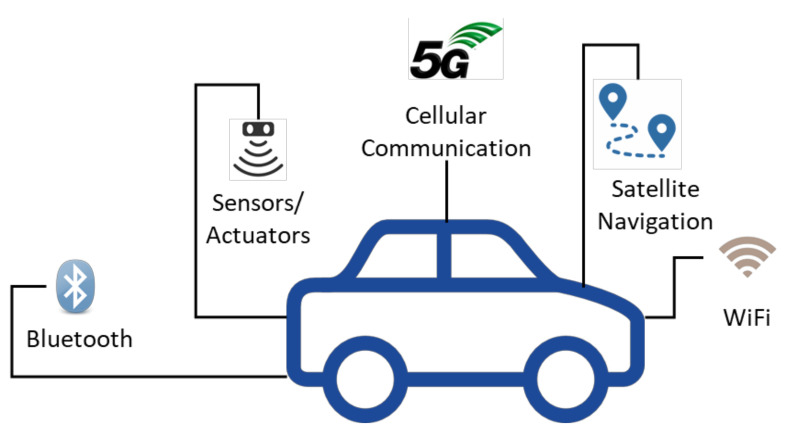
A typical smart vehicle (SV).

**Figure 2 sensors-21-03785-f002:**
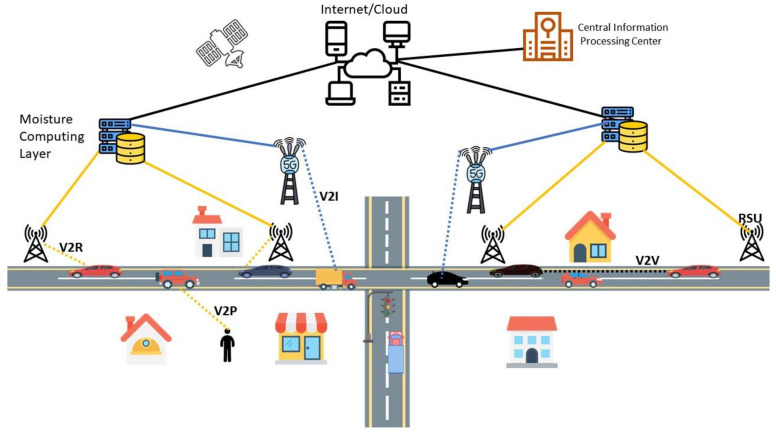
The proposed moisture computing Internet of vehicle (IoV) communication architecture.

**Figure 3 sensors-21-03785-f003:**
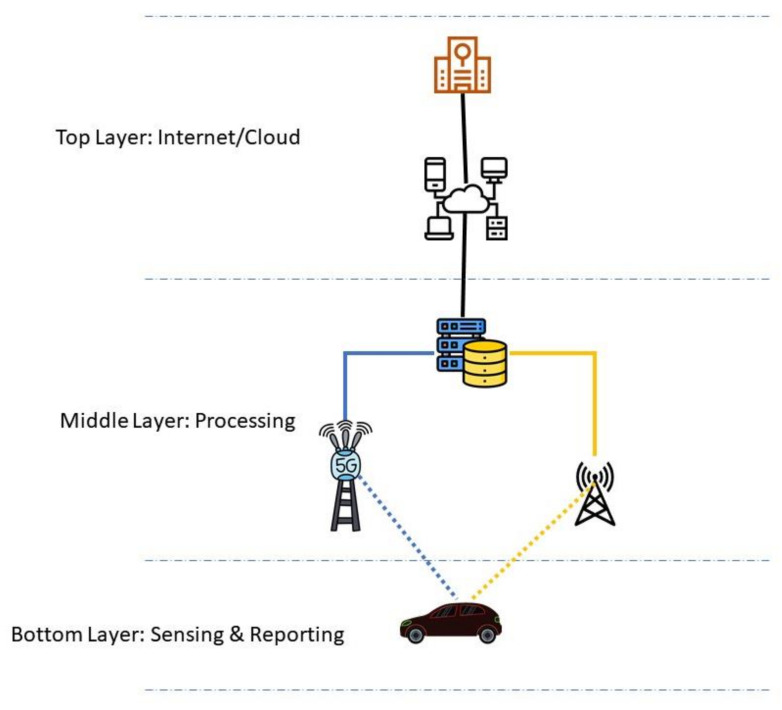
An overview of the proposed three-layer communication architecture.

**Figure 4 sensors-21-03785-f004:**
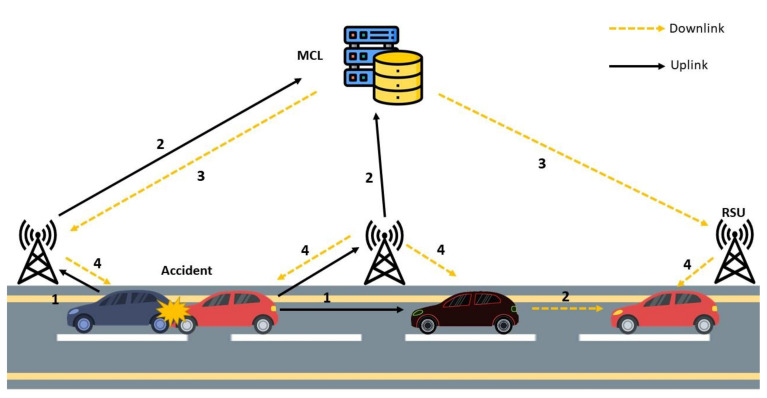
A communication scenario in the sensing and reporting layer.

**Figure 5 sensors-21-03785-f005:**
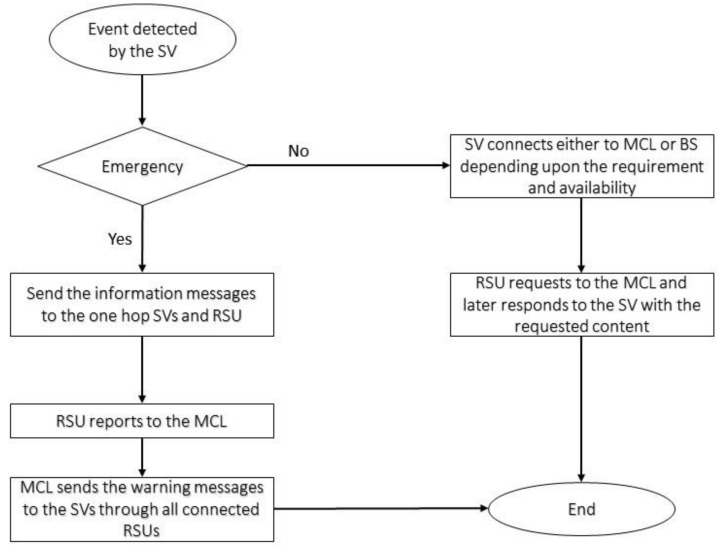
The interaction flowchart of the sensing layer, reporting layer, and processing layer.

**Figure 6 sensors-21-03785-f006:**
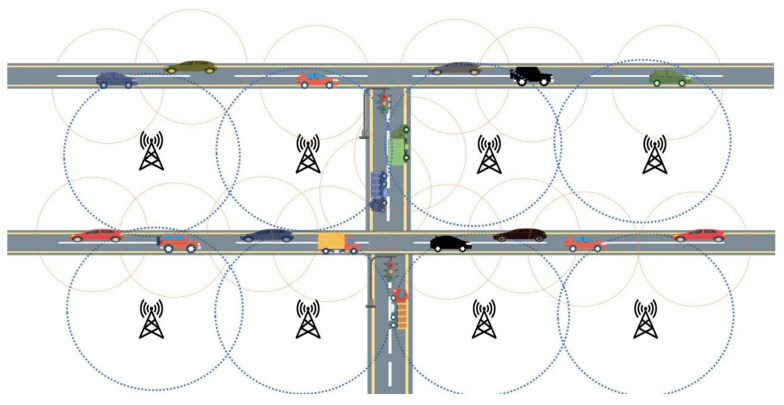
An urban/semi-urban environment, with communication ranges of SVs and road side units (RSUs).

**Figure 7 sensors-21-03785-f007:**
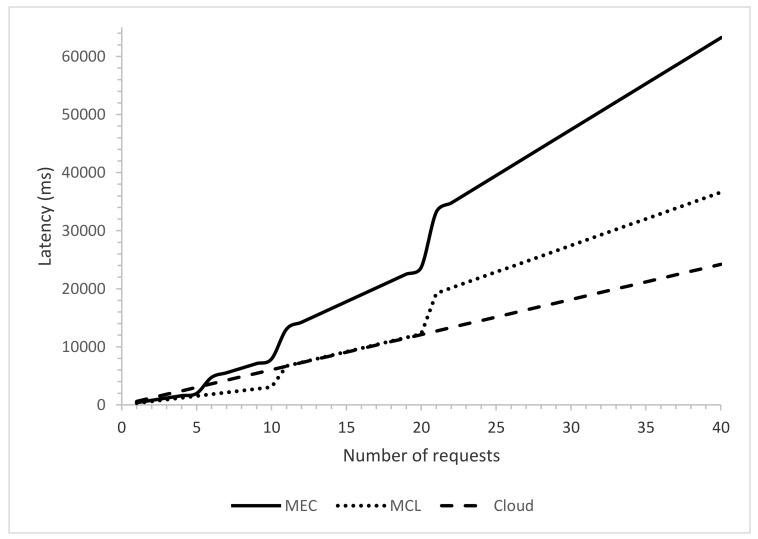
Latency against number of requests.

**Figure 8 sensors-21-03785-f008:**
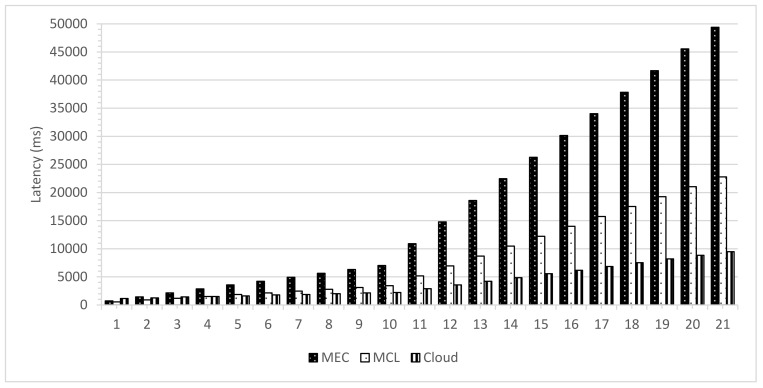
Latency with variable SVs and variable RSUs.

**Figure 9 sensors-21-03785-f009:**
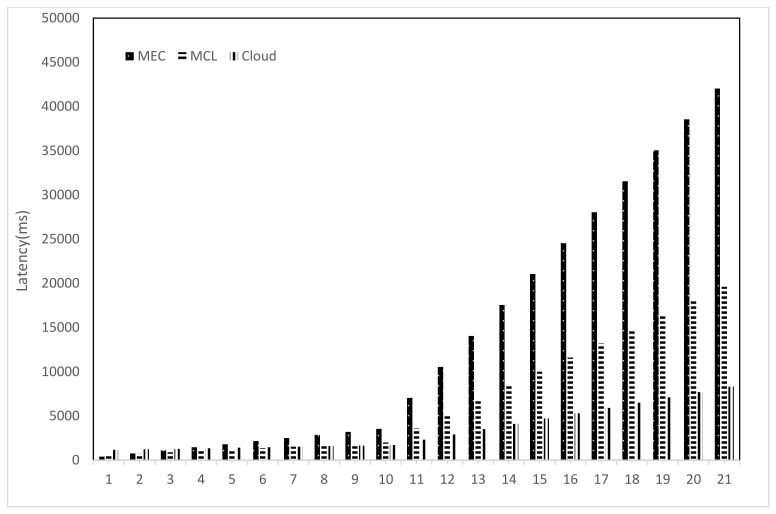
Latency with variable SVs and fixed RSUs.

**Figure 10 sensors-21-03785-f010:**
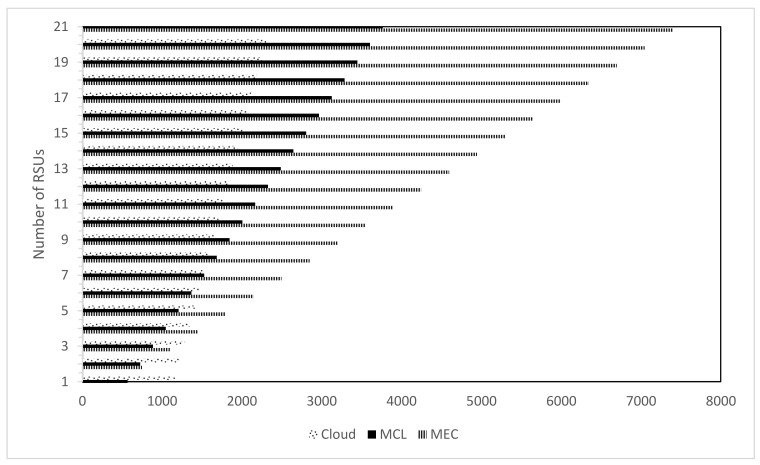
Network Latency with Fixed SVs and variable RSUs.

**Figure 11 sensors-21-03785-f011:**
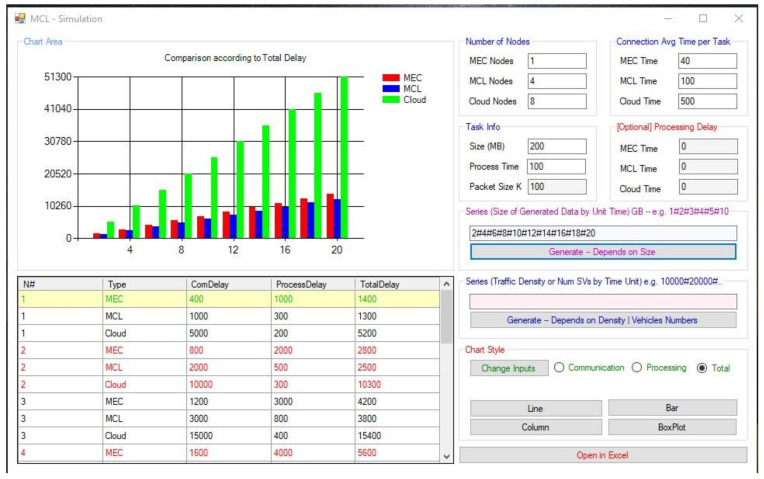
The Computational performance simulation tool.

**Figure 12 sensors-21-03785-f012:**
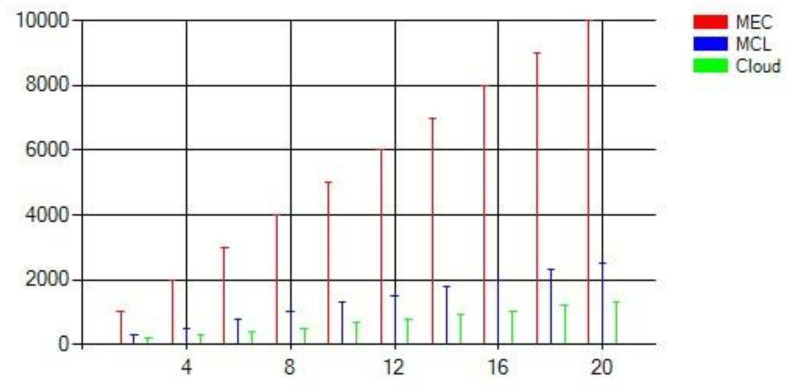
The computation time of MEC, the MCL, and Cloud.

**Figure 13 sensors-21-03785-f013:**
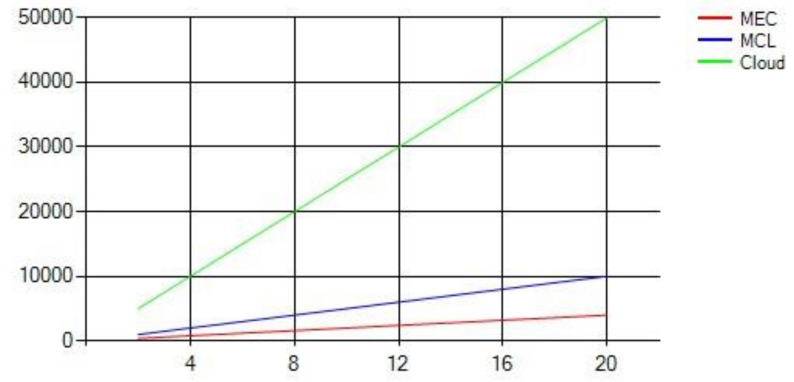
The communication delay of MEC, the MCL, and Cloud.

**Figure 14 sensors-21-03785-f014:**
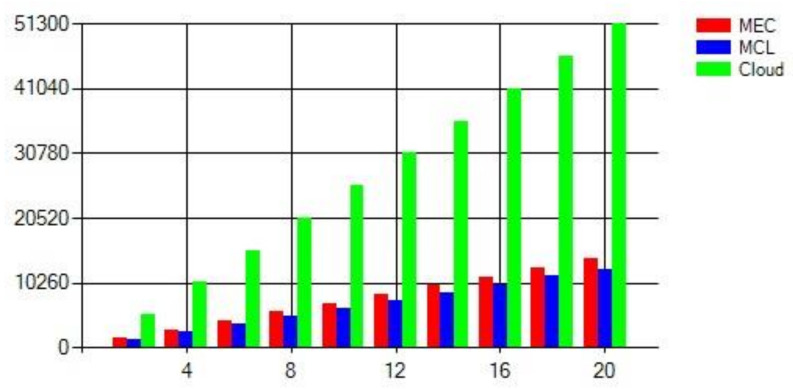
The total processing time of the MCL, MEC, and Cloud.

**Table 1 sensors-21-03785-t001:** Comparison of our proposed architecture with the existing architectures.

Related Studies	Distributed/Central/Hybrid	Type of Approach	Delay/ Latency	Domain of Application	Smart City Support	Cost Effectiveness	Density	Simulation/Analysis
[21]	Distributed	Multilayer-Fog Computing	×	×	✓	×	×	×
[22]	Distributed	Clustering Technique	✓	✓	×	×	×	✓
[23]	Distributed	Mobile Edge Computing	✓	Traffic safety and travel convenience	✓	×	✓	✓
[24]	Central	Cloud Computing	✓	Safety	×	✓	×	✓
[26]	Distributed	Mobile Edge Computing	✓	Immersive Vehicular Applications	×	✓	×	✓
[27]	Distributed	Co-operative Communication	✓	Road Safety and Security	×	×	×	✓
[28]	Distributed	Sensing as a Service Model	×	×	×	×	×	✓
[25]	Distributed	Agent-Based Social Computing	✓	×	×	×	×	✓
[17]	Hybrid	SDN and Mobile Edge Computing	✓	Delay Sensitive Applications	✓	×	×	✓
[10]	Hybrid	SDN and Fog Computing	✓	×	×	×	×	✓
[19]	Distributed	Fog Computing	×	×	×	×	×	✓
[20]	Distributed	Edge Computing	✓	Multimedia- Based	×	×	✓	✓
Our Proposed Architecture	Distributed	Moisture Computing	✓	Emergency and Content-Based	✓	✓	✓	✓

**Table 2 sensors-21-03785-t002:** The required infrastructure for the proposed moisture computing (MC) architecture.

Component	Main Tasks	Number of Units per Area
RSUs	Receive and relay messages	1–5
SVs	Send alerts and access services	1–Max (road capacity)
MCL	Receives requests and performs analytics	1
BSs	Receive requests from the SVs and provide communication via cellular network	1–3
Cloud	Provides Internet connectivity, data storage, data processing, and other critical services	1
CIPC	Possesses the analytics tools and can be used to provide smart services to the SVs	1

**Table 3 sensors-21-03785-t003:** The key criteria of mobile edge computing (MEC), cloud computing, and MC architecture.

Criterion	Cloud Computing Architecture	MEC Architecture	Moisture Computing Architecture
Architecture Type	Centralized	Distributed	Distributed
Latency with high density	High	High	Low
Latency for sparse networks	High	Low	Low
Near real-time interaction	No	Yes	Partial
Server Location	Tens of kilometers away from the end devices	Collocated with BSs or RSUs/Edge of the network	Middle or towards the edge of the network
Context Awareness	No	Yes	Yes
Cost for dense networks	High	High	Medium
Cost for sparse networks	High	Low	Low

**Table 4 sensors-21-03785-t004:** Analysis parameters used for comparing the moisture computing layer (MCL), cloud computing, and mobile edge computing architecture; ms = milliseconds, MB = megabytes.

Parameter	Value	Parameter	Value
Number of RSUs	1–20	Propagation Delay: SV-RSU	40 ms
Number of SVs	1–120	Propagation Delay: RSU-MCL	100 ms
Number of Servers in MEC	1	Propagation Delay: RSU-Cloud	500 ms
Number of Servers in the MCL	2–4	Per Task Processing Time at MEC	350 ms
Number of Servers in Cloud	5–8	Per Task Processing Time in the MCL	150 ms
Data per Task	500 MB	Per Task Processing Time at Cloud	50 ms
SV Sensing and Transmission Time	10 ms		

**Table 5 sensors-21-03785-t005:** Settings of implementation-based simulation for our comparison scenario; ms = milliseconds, KB = kilobytes, MB = megabytes.

Parameter	Value
Task size	200 MB
Packet size	100 KB
Server nodes in MEC	1
Server nodes in the MCL	4
Server nodes in Cloud	8
Server load balancing	Round robin
Task processing time per server node	100 ms per unit time
MEC processing delay (ms)	800
MCL processing delay (ms)	200
Cloud processing delay (ms)	100
RSU–MEC propagation delay (ms)	40
RSU–MCL propagation delay (ms)	100
RSU–Cloud propagation delay (ms)	500

## Data Availability

GitHub repository (https://github.com/anamoun/mcl.git, accessed on 28 May 2021).

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
