# Peer review of "Moisture Computing-Based Internet of Vehicles (IoV) Architecture for Smart Cities"

_sensors, 2021, doi:10.3390/s21113785_

Round 1

Reviewer 1 Report

The paper results interesting; the reviewer recognizes the efforts of the authors. In this sense, this reviewer has the following comments:

1) In telecommunication area, MC is widely used for Monte-Carlo-based methods. In this sense, it could be difficult to relate to Moisture Computing. Maybe, you should to propose a new term.

2) It is necessary to improve the quality of figures; e.g., Figures 2, 4, 5, among others. They can improved by using vectorial images.

Reviewer 2 Report

The authors provide an overview of different levels of computing in V2X networks and introduce their own intermediate layer. 

Sadly, some of the underlying assumptions seem to no longer be the right ones or need to be fleshed out more. For example, 5G rollouts will significantly increase the connected autonomous cars, but with a centralized and well-timed environment with guarantees. The level of MCL between "cloud" and roadside seems arbitrary and matches already existing 5G environments with significant computing in network (COIN). Lastly, the performance evaluation seems to be entirely based on static assumptions and there is a significant lack of detail outlining why performance degradations will happen.

Additional comments:

General

Please integrate a discussion on how MCL is different and enhances SDN/NFV/MEC in the 5G domain, as several approaches already discussed in the literature exist. For example, network slicing with different QoS levels per slice, including URLLC has already been discussed in larger extends. While MCL fits into this overall, it is not clear how it enhances on the existing network provider’s resource deployments that will be close to the RAN.

Research / Results

All evaluations are with static values, there are no dynamics, or at least they are not clarified. With a *vehicular* environment, the static evaluation might be less insightful than at least providing upper bounds through simulation.

Reviewer 3 Report

It should be interesting as the article proposed a new computation concept of "moisture computing". In order to promote this concept, the feasibility of moisture computing should be described in detail in the article. However how the authors got the fig 7 to 10 was not given the detail.  I suggest the authors provide more detail information about how you implemented this concept.  Also, please check the language problems in the lines 24,57,67,69-73,74,78,144,160-163,165 and so on.

Reviewer 4 Report

1. The authors should consider more recent research done in the field of their study (especially in the years 2019 and 2020 onwards). The reader may want to see how this work differs from other previous works. Moreover, the authors should clearly describe related work in more detail, contrasting the limitations of the related works.
2. This manuscript title is "Moisture Computing-based". However, the manuscript missed presenting the computing performance.
3. The proposed processes should be revised in a more formal pseudocode template. Moreover, the authors should include more technical details and explanations.
4. The comparison to other improved schemes (within the last 3 years) is required. The authors should provide enough proof to convince the reader of superiority of the proposed schemes over the existing works.
5. Some parameters and their values are unknown. It would be better to show all these parameters and explain the reason for those numbers in the table.
6. There is no discussion of user requirements, technological options and support for the decisions made at the design in this manuscript.
7. The conclusion should be rewritten to clarify your contribution. Please points out some insufficiency and limitation that needs further improvements in the conclusion. Moreover, formats of reference list lack consistency.

Round 2

Reviewer 2 Report

The authors have made a significant effort and improved the manuscript considerably. The revised version seems ready for publication, but the linked external files are not available in the git repository, so I chose to select a minor revision to ensure that the readers and reviewers will be able to inspect those as well.

Author Response

The authors have made a significant effort and improved the manuscript considerably. The revised version seems ready for publication, but the linked external files are not available in the git repository, so I chose to select a minor revision to ensure that the readers and reviewers will be able to inspect those as well.

Response: Thank you very much for your feedback. We have fixed the issue with the external repository. It is now accessible at the following link: https://github.com/anamoun/mcl

We gave also proofread the manuscript to fix the minor issues.

Reviewer 3 Report

The authors have proposed a good concept of moisture computing for the internet of vehicles. The reviewer hopes your authors continue to promote and implement this concept in the near future study.  

Author Response

The authors have proposed a good concept of moisture computing for the internet of vehicles. The reviewer hopes your authors continue to promote and implement this concept in the near future study.  

Response: Thank you very much for your feedback and appreciation. We will surely try to implement the proposed concept in real-time scenarios to validate it. 

Reviewer 4 Report

This paper has edited and revised according to the reviewer's suggestions.

Author Response

This paper has edited and revised according to the reviewer's suggestions.

Response: Thank you very much for your feedback and appreciation.